

# Development of a New Centralized Data Acquisition System for Seismic Exploration

Feng Guo[1], Qisheng Zhang[1], Qimao Zhang[2], Wenhao Li[1], Yueyun Luo[1], Yuefeng Niu[1], Shuaiqing Qiao[1]

[1]School of Geophysics and Information Technology, China University of Geosciences Beijing, No.29 Xueyuan Road, Haidian District, Beijing, 100083, China.
[2]Institute of Electronics, Chinese Academy of Science, No.19 North 4th Ring Road West, Haidian District, Beijing, 100190, China.

*Corresponding to*: Qisheng Zhang. zqs@cugb.edu.cn

**Abstract.** Seismic exploration equipment has developed rapidly over the past few decades. One such piece of equipment is a centralized seismograph, which plays an important role in engineering, so improving its performance is of great scientific significance. In this research, the core part of general seismic data acquisition devices is packaged to develop a centralized seismic data acquisition system (Named as CUGB-CS48DAS) that has independent operating ability and high scalability, which can be used for seismic exploration in varies engineering usage. Furthermore, by extending and modifying the acquisition circuit and corresponding software, the function of electrical method data acquisition has also been achieved. Thus, the proposed CUGB-CS48DAS makes it possible for joint exploration of seismic and electrical data in a single acquisition station, which is implicitly of great convenience in engineering prospecting. The low-power-consumption computer of the system comprises a 24-bit $\Sigma$-$\Delta$ modulation A/D converter and 48 sampling channels with an optional sampling rate of 50 Hz to 64 kHz, dynamic range $\geq$ 120 dB, synchronization accuracy better than 200 ns. With regard to the host computer, the architecture of the control software is smart, and it can integrate the multiple functions of data acquisition, preprocessing, and self-testing; clear interfaces reduce the complexity of development and migration. Field tests was implemented to prove that the system is stable and convenient to use, and the performance could meet the demand of high-precision joint exploration.

**Keywords:** Centralized Seismograph, Seismic Exploration, Electrical Prospecting, Data Acquisition, NB-IoT

**1 Introduction**

The development of seismic instruments has relied heavily on the continuous development of seismic data acquisition methods (S. Q. Qiao et al., 2018). Over the last 80 years, seismic exploration instruments have undergone five major developments: the electron tube (optical spot recorders), transistors (analog tape recorders), conventional digital seismographs (digital tape recorders), 16-bit telemetry seismographs, and 24-bit telemetry seismographs (K. Z. Song et al., 2012). At the beginning of

the 21st century, ION and Sercel launched the IV system and the 400 series of fully digital seismographs, respectively, which were recognized as sixth-generation seismographs in the industry. The development of seismic instruments cannot be isolated from the continuous development of seismic-data acquisition methods, combined with the latest technology available at the



time. Currently, the development of seismographs is highly dependent on the progress of electronic technology (X. Zhao et al., 2015), computer science, seismic exploration methods, intelligent control, network technology (G. Liang and W. Li, 2016), signal processing, and other disciplines (Q. S. Zhang et al., 2012), this development mutually reinforces the progress of the aforementioned disciplines.

The change from analog circuits to digital circuits has significantly increased the sampling accuracy, dynamic range, passband width, and number of data channels (S. Nakagawa, 2011). In terms of hardware circuitry, the presence of an analog–digital conversion acquisition chip and the high-performance conditioning filter circuit allows for a theoretical sampling precision of up to 24 bits and instantaneous dynamic range of more than 120 dB. In terms of software, the automation of instrument

operation allows for simple and convenient data processing.

The integrated framework of the centralized seismograph is usually fixed after entering the digital stage. Consider for example engineering seismographs such as SmartSeisST and ES-3000 from Laurel Technologies, whose major structure comprises a sensor (detector) receiving end, a signal conditioning circuit, an analog–digital conversion circuit, a control logic, and a

human–computer interface (HCI) along the direction of data flow transmission (S. Mazza et al., 2012). Therefore, the major structure of the centralized seismograph is relatively stable. The difference between various models lies in the auxiliary equipment (printer, built-in GPS, etc.), the channel number of the acquisition circuit, sample performance parameters, or instrument interface (Z. Q. Wu et al., 2011). Hence, we can separately package the conditioning circuit, the AD conversion circuit, the logic controller, and the software into a fixed core acquisition system. In this system, hardware and software can

be customized and extended to acquire data via different seismic exploration methods, thus saving plenty of resources in the development of the instrument.

As is known, ambiguity problem is hard to overcome with single geophysical method. Thus, an idea of joint inversion with multiple geophysical methods was introduced and has been proved to be effective. Under this circumstance, single instrument

which can achieve joint seismic and electrical data acquisition is essential. In 2013, the Italian company named P.A.S.I launched the combined imaging system 16SG24-N for engineering seismic and high-density electrical acquisition. And the KMS-850 from KMS company can achieve data acquisition of microseismic and multi-functional electromagnetic methods (Q.S. Zhang et al, 2012).  However, the 16SG24-N has already been out of production a couple of years ago and was altered by 2 other instruments with single function. Therefore, the idea came up to develop a combined acquisition system with multi-

function which is easy to use.

With this idea, a scalable data acquisition system (Named as CUGB-CS48DAS, shown as Figure 1) is designed on the basis of the common PC104. The CUGB-CS48DAS data acquisition system can be used for seismic exploration as well as electrical prospecting. It has good application effects in engineering geology, mineral geology and energy geology, and is suitable for





exploration tasks in coalfields, petroleum, minerals, earthquakes and urban construction, etc. In particular, with the introduction of the technology named Narrow Band Internet of Things (NB-IoT), it will be simpler to realize multi-system networking, which provides great convenience for unified monitoring and management of the acquisition instruments during the exploration.

This paper focuses on describing the hardware and software architecture of the CUGB-CS48DAS system, as well as its implementation mechanism. The feasibility of this system is then demonstrated by using a specific case.

## 2 Structure of the Acquisition System

The development of the acquisition system generally involves hardware circuit design and control software development (D. Marjanovic et al., 2015). The CUGB-CS48DAS system structure is shown in Figure 2. Basic performance parameters and functions of the acquisition system depend on the hardware circuit, which is essential for ensure the quality of the acquired data. The operating state of the overall system is coordinated by the software, which is subject to certain fault tolerance and more flexible than the hardware circuit. It is required to encapsulate operations near hardware bottom and provide flexible and safe interfaces, thus both ensuring the system stability and improving its flexibility and portability.

### 2.1 Hardware Architecture

The acquisition system consists of a host control module, a power module, an acquisition control module, and an acquisition unit array, the CUGB-CS48DAS system structure is shown in Figure 2.

The host control module can implement system status control, data concentration, and human–computer interaction. The module mainly adopts a PC104 (model)-based embedded system with a Windows XP Embedded operating system and software written and compiled in MSVC. The host control module itself has two USB 2.0 interfaces, four RS232 serial ports, one 100 M network port, and one screen LVDS port. Two of the four serial ports are used by the host computer to control the operating state of the other modules. The other two ports are reserved for functional extensions. The 100 M network is used for transmitting acquired seismic data. The LVDS data cables are connected to an external LCD screen with adjustable backlight brightness. The USB interfaces are connected to a Bluetooth peripheral as the input device of the instrument.

The power module, which is shown in Figure 3(b), is used to supply power to the system, as well as to monitor all module operating states and to condition acquired synchronizing signals, etc. The module consists of an MCU circuit, a level switch circuit, and a synchronizing trigger circuit. The MCU circuit mainly consists of a C8051F320 microcontroller and related peripheral circuits. It can supply power to different system modules by strobing for relays of the level switch circuit. It also receives operating parameters of circuits via its own AD converter to monitor the operating states of modules. The level switch



circuit can convert the external 12 V power into operating voltages required for the sub-modules of the acquisition system. The design needs to provide level switch stability to ensure normal operation of the acquisition system and accuracy of data acquisition. The synchronizing trigger circuit can isolate the acquired synchronizing signals, shape the signals, and then transmit them to the acquisition control module and the acquisition unit array to ensure temporal synchronization of channel

data.

The acquisition control module implements logic control of the acquisition unit array and transmission of acquired data, and details of the acquisition board can be found in Figure 3(C). The modular circuit mainly consists of a field-programmable gate array (FPGA) chip (EP3C25F324I7N) and a 100 M network chip (DM9000). The module function mainly relies on the FPGA.

A soft CPU is set up on the FPGA by using system-on-a-programmable-chip (SoPC) technology to communicate with other modules. The circuit module written in VHDL can provide data transmission between modules. And the network chip DM9000 was introduced in order to realize the communication between upper computer (PC104) and acquisition module.

The acquisition unit array comprises multiple acquisition units, each of which consists of one host controller FPGA and one

group of AD conversion circuits (ADS1274). The controller can achieve control of conversion circuits and communication for the acquisition control system by setting up an SoPC on the FPGA. The AD conversion circuit mainly comprises three 4-channel AD converters (ADS1274) and relevant conditioning circuits.

The circuit boards internal include a main control board, a power supply board, and four acquisition boards, which are inserted

in the slot of an aluminium alloy frame as is shown in the Figure3(a). Except for the main control board, all other circuit boards can be removed easily from the slot, thus providing conveniences to subsequent upgrade and maintenance. A shielding layer is placed between the power board and the acquisition board to reduce the impact of electromagnetic radiation on the acquired signal.

## 2.2 System Software Description

The system software runs under the embedded Windows XP operating system to balance the demand of a graphic display interface and the consideration of limit performance of PC104. The host computer software is designed on the basis of the MFC provided by Microsoft. The software framework in this study is shown in Figure 4. The system software architecture comprises an interactive interface, background program, and embedded system. The drawing module, methods manager, and document manager of the interactive interface can be adjusted or replaced depending on actual method requirements. The

process controller of the background program is the core of the software that provides a functional interface for interface software development.



The process controller is mainly designed to enable the host computer control system to communicate with the power supply and the acquisition control system, and it can complete a full set of acquisition processes based on different methods and monitor the operating states of the instrument parts. The process controller comprises three basic units (a power control unit, a data acquisition control unit, and a data transmission unit) to provide services. Each unit consists of three layers, i.e., the

implementation layer, protocol layer, and control layer. The implementation layer facilitates communication between the software and underlying operating system. The protocol layer can parse data of the drive layer to find corresponding commands or data in the protocol table. The control layer, as a functional interface for the above two layers, can transmit messages corresponding to commands, so that each module can complete a series of tasks under the management of the process manager.

The exploration methods manager uniformly converts exploration methods selected on the HCI and relevant parameters to a data structure that can be identified between software modules and that can ensure parameter synchronization between modules.

The document manager can read and format raw data. Format templates for data storage are selected based on specific exploration methods. For example, seismic exploration methods are converted to SEG-2, SEG-Y, and other formats.

The drawing module is used for the graphical display of data. It is designed to manage acquisition parameters and actual data blocks, as well as handle graphic display operations for users.

## 3 Core Technology for Acquisition System

It is critical for the overall acquisition system to achieve complete, stable, and portable acquisition workflow. In the acquisition

process, the process controller of the host control module plays an important role. Communication protocols and workflows between modules are defined by the process controller, which determines the operating state of the overall system. The process controller also acts as the hardware emulation layer in the aspect of architecture development, providing the software interface of the instrument with a complete application interface, which is essential for the overall acquisition system to possess high project portability.

### 3.1 Packaging Design of Process Controller

The packaging design of a process controller is shown in Figure 5. The process controller can provide a data interface, functional interface, and message interface for the host control system software. Implementing data communication and parameter setting as a pure software program.

When the process controller works, multiple threads and tasks will start simultaneously and data as well as message communication will be required between the functional layers and between the sub-modules. In order to ensure normal





software operation, communication between threads under the same sub-module should be regulated via a synchronizer and communication between functional layers should be regulated via a message stack. Part of the message stack is packaged as an interface to facilitate flexibility of the process controller. The two message stacks in Figure 5 are of different usage. The one connected to the web server and data processor is related to data upload, which is achieved through the network due to the
large data volume, represents the network data receiver based on UDP protocol in the upper computer software. Another message stack is used to transmit the command and control messages through UART. These two types of message stacks work separately to complete the whole data acquisition process.

The data interface partitions part of the memory area in service as the hard disk (HD) area by using RAM Disk. The user
interface is in essence used for the mapping of cache files in the memory area. It can ensure high access speed of cache files and security in file sharing.

The functional interface mainly provides performance functions and public variables for operating the process controller, such as the sub-module state inquiry and configuration.

The message interface consists of a process message table and message allocation threads. The process message table is subject to the fixed priority setting, to collect and distribute messages of the process controller at work. Generally, error messages are at top priority, followed by network data–related messages, with a view to ensure the instrument stability and high transmission rate of acquired data. The message sent from the serial port is at low priority, followed by a return message. It involves
operating the relay of the lower computer and other hardware configured with a low speed (≥100 ms), so the requirement for response time is low. Each sub-module has its own message stack. Communication between sub-modules and that between sub-module and the process controller are assigned via a message interface at the upper level.

## 3.2 System Acquisition Process Design

Single data acquisition is the most basic unit for system acquisition. Other acquisitions of the system are achieved by adjusting
or calling the single data acquisition.

For each acquisition, the power of the acquisition control module and the acquisition unit array has to be turned on, wait for the termination of initialization, and then configure acquisition parameters for the acquisition control module. For formal acquisition, a command shall be sent to the acquisition control module to make it ready and return a message to the host
computer; the host computer then enters the data receiving mode. The acquisition control module transmits data to the host computer via the network. After the host control system receives network data packages, the data transfer unit will receive and process the data package. After the acquisition is completed, the data transfer unit will enter the preprocessing mode to collate and cache the data. Next, the acquisition mode is determined. The data will be processed by self-test procedures to obtain a



self-test report when the instrument self-test is conducted. For normal acquisition, the data will be sent to the document manager and the drawing unit for processing. When the continuous acquisition is completed, the process controller will reset and turn off the acquisition control module and the acquisition unit array.

In the acquisition system, the actual data transfer rate and the preset sampling rate share a linear relationship. The effective data transfer rate should reach 16 Mbps for normal operation. Test results have shown that the data transfer rate between the acquisition control module and the data acquisition unit array can reach 100 Mbps and that between the acquisition control module and the host control module can reach 90 Mbps, while the data receiving rate of the host control module is generally about 16 Mbps.

Data receiving in the host control module is a bottleneck for the entire system. Hence, the thread concurrency should be minimized in the single acquisition design in order to improve the data transfer rate and stability of the system. In particular, data receiving is separated from data feedback on the HMI, while the timing query of the message stack is used in the drawing part to confirm data acquisition progress and display it on the interface. After the abovementioned processing, the host control

module can attain a data receiving rate of up to 90 Mbps, which can meet the demand of normal operation (16 Mbps).

### 3.3 System Data Transmission Technology

For the convenience of indoor data processing and interpretation of seismic data, field data collection accuracy has to be high, and the bit error rate has to be low. To ensure the accuracy of data, the collected seismic data need to be encoded and then sent from the Acquisition Unit to the Acquisition Control Module. The transmission coding method of CUGB- CS48DAS uses

Manchester Encoding, which has the advantage of being able to extract the synchronization clock from the signal more easily.

The encoding process is completed in the Acquisition Unit, and the encoding module is written in VHDL. The parallel data in bytes are converted into one-bit serial data which are encoded in the form of Manchester Encoding. During the encoding process, binary "0" is converted to "01", and binary "1" is converted to "10". Since the receiving end needs to align the valid

edge with a certain period of time to achieve constant synchronization, a preamble must be sent before sending one frame of data. Then the encoded data are transmitted via LVDS which takes advantage of low noise and low power consumption

The Acquisition Control Module receives the data from Acquisition Unit through the LVDS interface. After receiving the preamble, the receiving end will complete the clock synchronization in several data cycles. The decoding module, which is

also written in VHDL, uses a clock 8 times that of the data speed to detect the data jumps during the effective jumping time. Once the data have an early or a delayed transition, the decoding module automatically aligns with the new edge. In this way, the data can be allowed to shake to a large extent without causing bit errors. Besides, the decoding module also cope with



synchronized header determination and byte synchronization, etc. Eventually, the data are saved in the SDRAM and they can be acquired by the Host Control Module through 100 M network.

## 4 Use Cases

As known from the system description, the overall system can independently perform basic functions such as instrument self-testing and data storage by using the process controller and the core hardware. Therefore, the process controller and core hardware can be packaged as a whole to provide technical support for other acquisition systems with different functions.

The porting of the acquisition system is accomplished in three steps. (1) The acquisition systems should be able meet project performance requirements. (2) In terms of hardware, the circuit board size and the layout of device connectors and modules should be adjusted. (3) In terms of software, the process controller interfaces should be compiled with the interactive interface, methods manager, document manager, and drawing module.

It is easy to implement this method technically. In terms of hardware, the logic control part is included in the FPGA chip with a generally fixed interface, and it can be easily upgraded. In terms of software, the process controller is as assumed to be an independent linkbase. The methods manager, the document manager, and the drawing module can use it for joint compilation. Even the scripting language is used to provide an interactive interface function, thus greatly enhancing system reusability.

The acquisition system has been verified in actual engineering applications. In addition, some cases are employed to demonstrate great scalability of the system.

### 4.1 Application of Centralized Seismic Exploration Instrument

CUGB-CS48DAS is an engineering seismic exploration instrument obtained by customizing certain functions of the core acquisition system. The monitoring mechanism of the power module and data acquisition mechanism of the acquisition control part are optimized. Nine common engineering seismic exploration methods such as coverage measurement, reflection, refraction, and surface wave are included in the method selector of the host control system. Some interfaces of the drawing module and the document manager are modified.

The instrument was used for a two-week seismic exploration in an area of Golmud City, Qinghai Province in 2013; the total measuring line length was 4.4 km, the seismic migration profile is shown in Figure 6. The exploration adopted the CDJ-Z/P60 geophone of the Chongqing Instrument Factory (technical parameters is listed in Appendix A) and explosives as a seismic source. The sampling rate was 4 kHz, with 2048 sampling points and 24 channels. Seismic surveying utilizes the coverage measurement method.



## 4.2 Realization of Distributed Seismic Exploration Instrument

CUGB-CS48DAS was designed to provide additional distributed acquisition station functions on the basis of 48 channels to
the original system and achieve data transfer and command control via network cables. Distributed control was added to the
acquisition control system on the hardware layer. In the host control system, the operating interface, the drawing module, and
the document manager were modified and 48 sampling channels were extended to 96, the system diagram is shown in Figure
7, connect CUGB-CS48DAS and Distributed Sampling Unit with PoE (Power Over Ethernet).

## 4.3 Extension of Electrical Prospecting

In August 2014, an extension of the electrical prospecting function was achieved on the basis of the CUGB-CS48DAS
seismograph. The extension of geophysical electrical prospecting is shown in Figure 8. An electrical prospecting controller
and an electrical transmitter were added to the original system, and sensors were changed to electrodes. Moreover,
modifications were implemented on the original acquisition boards designed only for seismic data acquisition to meet the
needs of electrical data acquisition as well. Mainly, the cut-off frequency of low-pass filter became optional up to 3kHz when
acquiring electrical data. Particularly, 50Hz power frequency filter circuits were added to realize a -82dB attenuation on that
frequency point. The host computer of the original system was directly connected to the electrical prospecting controller via
serial ports. The overall acquisition process can be controlled using the electrical prospecting controller which controls the
electrode channel configuration and operating state of the electrical transmitter based on host computer commands. The main
part of the electrical transmitter is the medium-power generators used for geophysical electrical prospecting of the measured
area, so that the measured objects show electrical characteristics (such as apparent resistivity and apparent polarizability). By
controlling the electrical transmitter and changing the frequency and waveform of the emission current, the measurements of
the induced polarization method in both frequency domain and time domain can be performed.

Field comparative experiment was then implemented at the suburban area of Beijing, in contrast with the high-density electrical
prospecting instrument of Chongqing Instrument Factory named DUK-2A. The pseudo-sections of apparent resistivity and
apparent polarizability is shown as Figure 9. As is indicated in Figure 9(a) and 8(b), the pseudo-sections of apparent resistivity
obtained by both instruments present high consistency, no matter in the trend that resistivity varies or the approximate
appearance and thickness of different layers. It could also be found that there are some differences between Figure 9(a) and (b)
presented as small scaled anomalies. This is because the two instruments were not placed at the exactly same location and
there were some errors introduced during inversion process. The Figure 9(c) and 8(d) are illustrations of apparent polarizability.
According to that, it is easy to find the contours of apparent polarizability acquired by proposed instrument are well
superimposed with that obtained using DUK-2A.


Therefore, it can be proved that the performance of CUGB-CS48DAS used as an electrical exploration receiver is in compliance with the requirements and has the application value in joint seismic and electrical prospecting.

## 4.4 Connection to the Internet of Things

Recently, the introduction of NB-IoT chip named BC95 enables connection to the Internet of things. Via UART interface, data exchange between the host control module and BC95 can be accomplished. Commercial network become available thanks to the internal GSM/GPRS module of BC95. When constructing urban underground space monitoring and natural disaster warning systems, we hope to deploy seismic acquisition stations in the areas covering the entire city to monitor the activity of underground media. However, it is not simple to build up the communication network covering a large area between acquisition stations and the central station if real-time quality control (QC) is expected. Taking the node seismographs systems as an example, the wireless data transmission methods between nodes and the central station are mainly: (1) a relay type transmission based on a multi-hop network, for example the scheme of WTU-508 system; (2) a wireless data transmission scheme based on a high-power directional Access Point (AP) connecting directly with the central station using the star topology. Among them, the multi-hop network has a short single-station interval, and the communication distance of the directional AP is basically no further than 5 km ideally. Therefore, there are plenty of limitations that cannot adapt to the complex environment of the city. The LTE-based data transmission method is also one of the strategies because extra communication network could be omitted, but the LTE network coverage capability is not satisfying. In fact, the base station of NB-IoT can increase the gain by 20dB compared with that of LTE (A. Adhikary et al., 2016), which means that stations can communicate normally in buildings and even underground garages and other places with obstacles. Another advantage of NB-IoT is that there is no need to build extra communication network. These advantages altogether provide great convenience for the layout of seismic instruments. In addition, with the cloud server of NB-IoT, it is also easy to establish a QC monitoring centre without setting up an instrument vehicle or a central station like that in traditional seismic survey.

As is illustrated in Figure 10, the proposed CUGB-CS48DAS acts as the terminal device and data from host control module are sent to the BC95 through the UART interface, then uploaded to the IoT open platform via NB-IoT base stations. The uploaded data could then be accessed by management device, which is usually a PC, after the processing of application services to distinguish the data according to their type. Conversely, commands can also be sent to the terminal device from management device to control the acquisition process.

As we can see, a screenshot of QC monitoring interface during a communication experiment we implemented is shown in Figure 11. Totally, 5 test points in the campus was selected to observe the network quality and communication speed. Therefore, we used the indicator Reference Signal Receiving Power (RSRP) to represent the signal intensity as is shown in Figure 11.



Generally speaking, the RSRP of 5 test points are around -85dBm which stand for good points of network quality. Under such circumstances, QC monitoring and acquisition control can be realized as is designed.

**5 Data Acquisition System Performance Indicators**

After conducting tests and analysis in the laboratory and in the field, the acquisition system developed has the following

performance indicators:

(1) An embedded Windows XP data acquisition interface that supports methods of surveying coverage measurement, reflection, and refraction. The system will be able to expand the collecting function of the geophysical electrical prospecting.

(2) Low-pass filtering: 0.8 FN (digital filter);

(3) Sampling rate: 50 Hz to 64 kHz;

(4) Word length: 24 bits;

(5) Power supply: DC 12 V;

(6) Crosstalk rejection ratio: ≥ 80 dB;

(7) Dynamic range: 120 dB at 0.4 kHz BW (1 kHz sampling rate);

(8) Total harmonic distortion: -105 dB;

(9) CMRR: >102 dB;

(10) Stop-band attenuation: >120 dB (above the Nyquist frequency);

(11) Noise (DC to 200 Hz): 450 nV RMS at 0 dB;

(12) Data transmission speed: 90 Mbps;

(13) Synchronization accuracy: < 200 ns;

(14) Operating temperature: -20°C to +70°C.

**Table 1: Comparison of parameters between 16SG24-N and CUGB-CS48DAS**

| | 16SG24-N | CUGB-CS48DAS |
|---|---|---|
| Display/inch | 10.6 (Touch screen) | 12.1 |
| Dimension/cm | 48×38×18 | 42×30×19 |
| Weight/ kg | 11 | 9.3 |
| Operation System | 32bit | 32bit |
| Number of Channels | 24 | 48 |
| Sampling Rate/kHz | 0.5~31.25 | 0.05~64 |
| Amplifier Gain | Automatic gain control | ×1/×10/×100 optional |
| Storage | 60GB HDD | 64GB CF card |





| Applications | Refraction & Shallow reflection & Surface waves & High-density electrical method | Refraction & Reflection & Surface waves & High-density electrical method |
|---|---|---|
| ADC | 24 bits, Sigma-Delta ADC | 24 bits, ADS1274 |
| Dynamic Range | >117 dB (instantaneous, @1ksps) | 120 dB (@1ksps, 0.4 kHz BW) |
| Operating Temperature/°C | -30~ +80 | -20~ +70 |

Comparing to the 16SG24-N, which represents the advanced level of combined system for seismic and electrical instrument, it can be found in Table 1 that the proposed CUGB-CS48DAS does not have obvious disadvantages and even outcompetes in some aspects like dynamic range and number of channels etc. Therefore, CUGB-CS48DAS is proved to be a competitive and

practical choice for joint seismic and electrical prospecting instrument with less budget.

**6 Conclusions**

In this study, a new type of centralized data acquisition system was designed, based on the SoPC technique. Through a research and development program, the following technical aspects were explored:

(1) In accordance with the framework and actual results, CUGB-CS48DAS is a complete and independent acquisition system. Practical applications and extension cases show the system has better scalability and portability, with scope for use in many applications. Further standardization of the application interfaces of the system is expected, along with a reduction in cost. We will attempt to apply this system to data acquisition tasks besides geophysical exploration.

(2) The seismic exploration data acquisition and processing technology using the SoPC technique as the master control core: this technology introduces the most recent SoPC technique into engineering seismic exploration, which reduces power consumption while improving seismograph resolution and synchronization accuracy. The system can also be flexibly embedded into the seismic exploration data processing algorithm in the form of hardware.

(3) 48-channel, 24-bit high-precision parallel AD conversion technique: the multi-channel seismic data acquisition performed in existing engineering seismic exploration uses only 1 AD converter, resulting in poor synchronization when carrying out time-shared data acquisition; whereas in this design, every channel uses a separate high-precision AD converter to provide synchronization precision.

(4) CUGB-CS48DAS can solve the ambiguity problem in geophysical prospecting and achieve joint geophysical seismic and electrical prospecting. The single method of present geophysical prospecting has an ambiguity problem and thus cannot be refined; nevertheless, this new centralized data collection system provides technical means to solve this problem while offering



useful explorations for joint geophysical seismic and electrical prospecting. However, the increase of synchronization precision and acquisition accuracy provides technical support for more refined seismic prospecting, yet because the increase of synchronization precision and acquisition accuracy has met the requirements of high-precision geophysical electrical prospecting, novel supporting equipment has been employed for joint geophysical seismic and electrical prospecting.

(5) The introduction of embedded Windows XP technology has improved the human–machine interactive interface of an engineering seismograph.

(6) The application of NB-IoT makes it possible and an easy way for networking between multiple CUGB-CS48DAS stations
10    and real-time QC monitoring.

**7 Funding Acknowledgement**

This work is supported by the Natural Science Foundation of China (No. 41574131 and No. 41204135), the National "863" Program of China (No. 2012AA061102 and No. 2012AA09A20102), the National Major Scientific Research Equipment Research Projects of China (No. ZDYZ2012-1-05-01), and the Fundamental Research Funds for the Central Universities of
15    China (No. 2652015213).

**Appendix A**

**Technical Parameters of Geophone CDJ-Z/P60**

| Model | Natural Frequency/Hz | Sensitivity (V/cm/s) | Coil Resistance (kΩ) | Internal Resistance (kΩ) |
|---|---|---|---|---|
| CDJ-Z/P60 | $60 \pm 5\%$ | $0.30 \pm 5\%$ | $1060 \pm 5\%$ | $800 \pm 5\%$ |
| Weight (g) | Damping Coefficient | Harmonic Distortion (%) | Coil Displacement (mm) | Operating Temp. (°C) |
| 6.3 | $0.6 \pm 10\%$ | $\leq 0.2$ | $\leq 2$ | -40~ +70 |

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

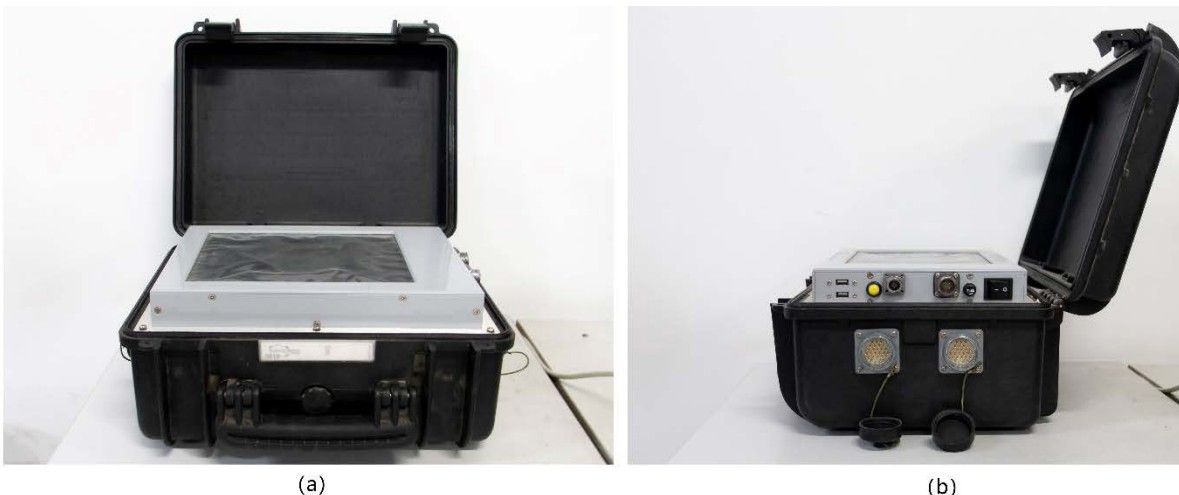

(a)                                                        (b)

**Figure 1: Photograph of realized CUGB-CS48DAS.**





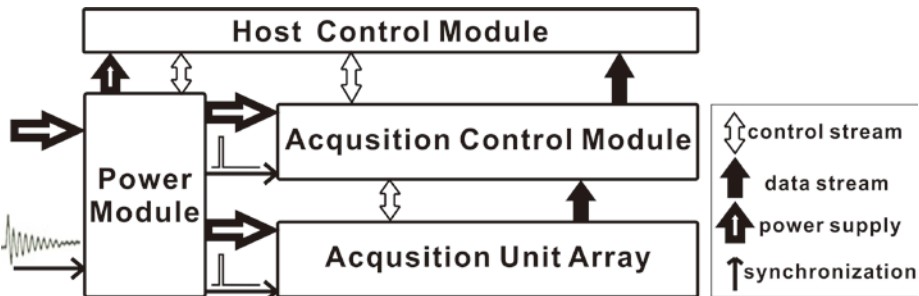

**Figure 2: Structure diagram of CUGB-CS48DAS.**

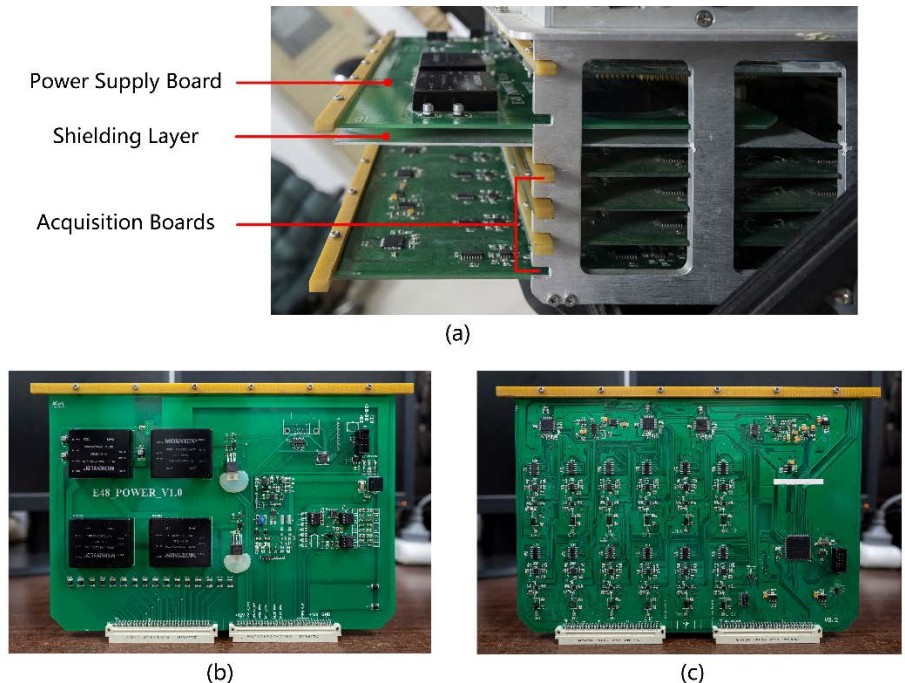

**Figure 3:** Internal structure of CUGB-CS48DAS and circuit boards. **(a) Packaging structure of circuit boards. (b) Realize of the Power Supply Board. (c) Realized of the Acquisition Board with 12 channels.**





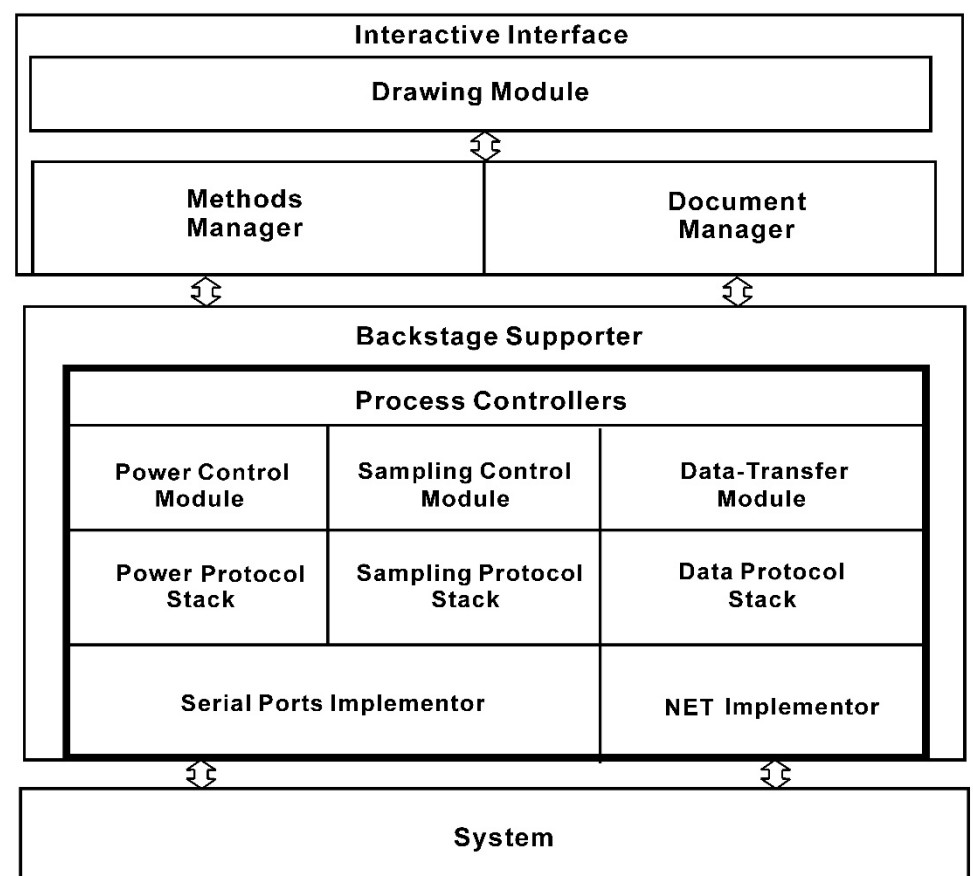

**Figure 4: Software architecture of host computer control system**

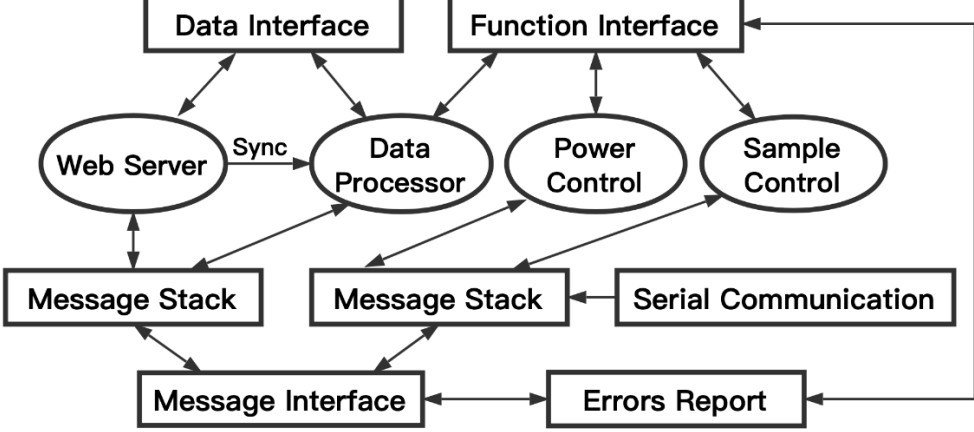

**Figure 5: The packaging design of a process controller**



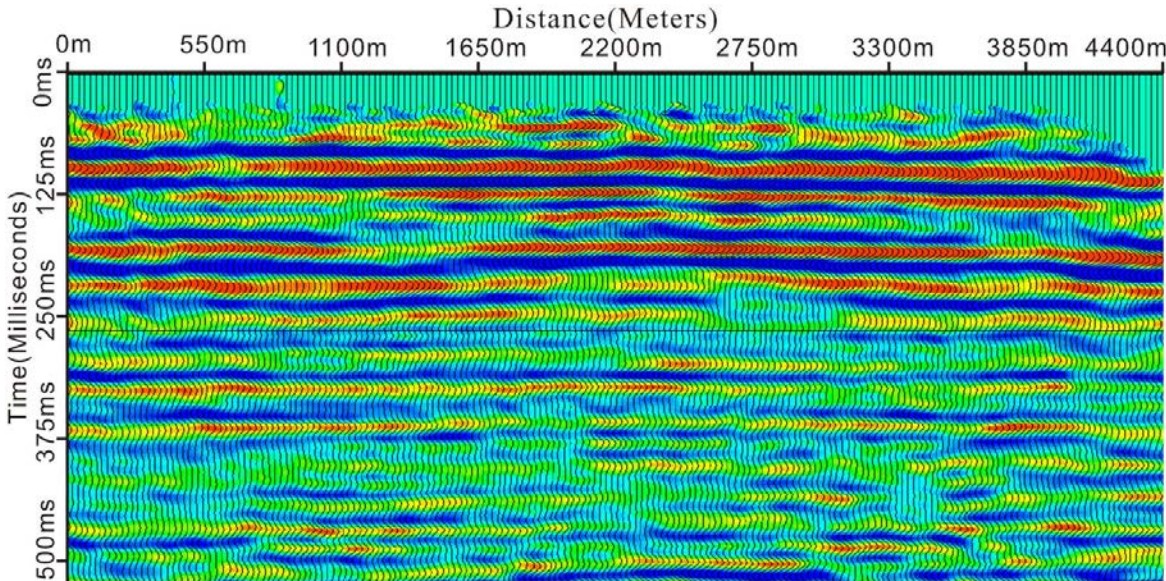

**Figure 6: Seismic migration profile**

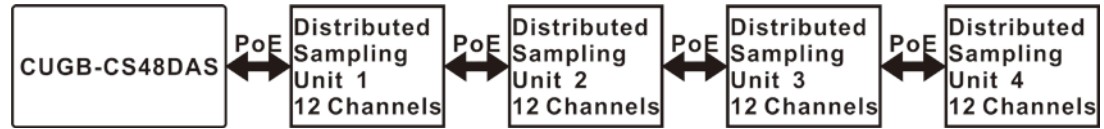

**Figure 7: Distributed and centralized seismograph**

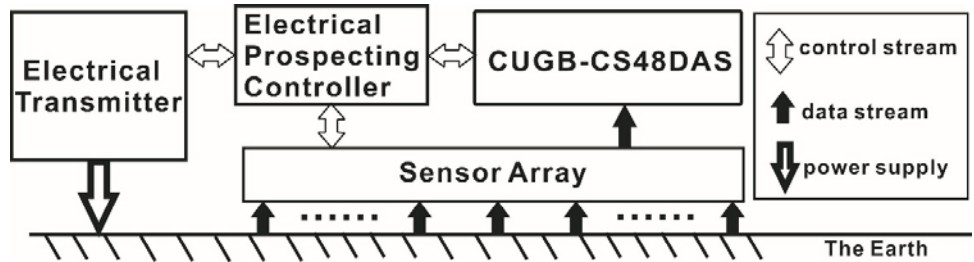

**Figure 8: Extension of geophysical electrical prospecting**



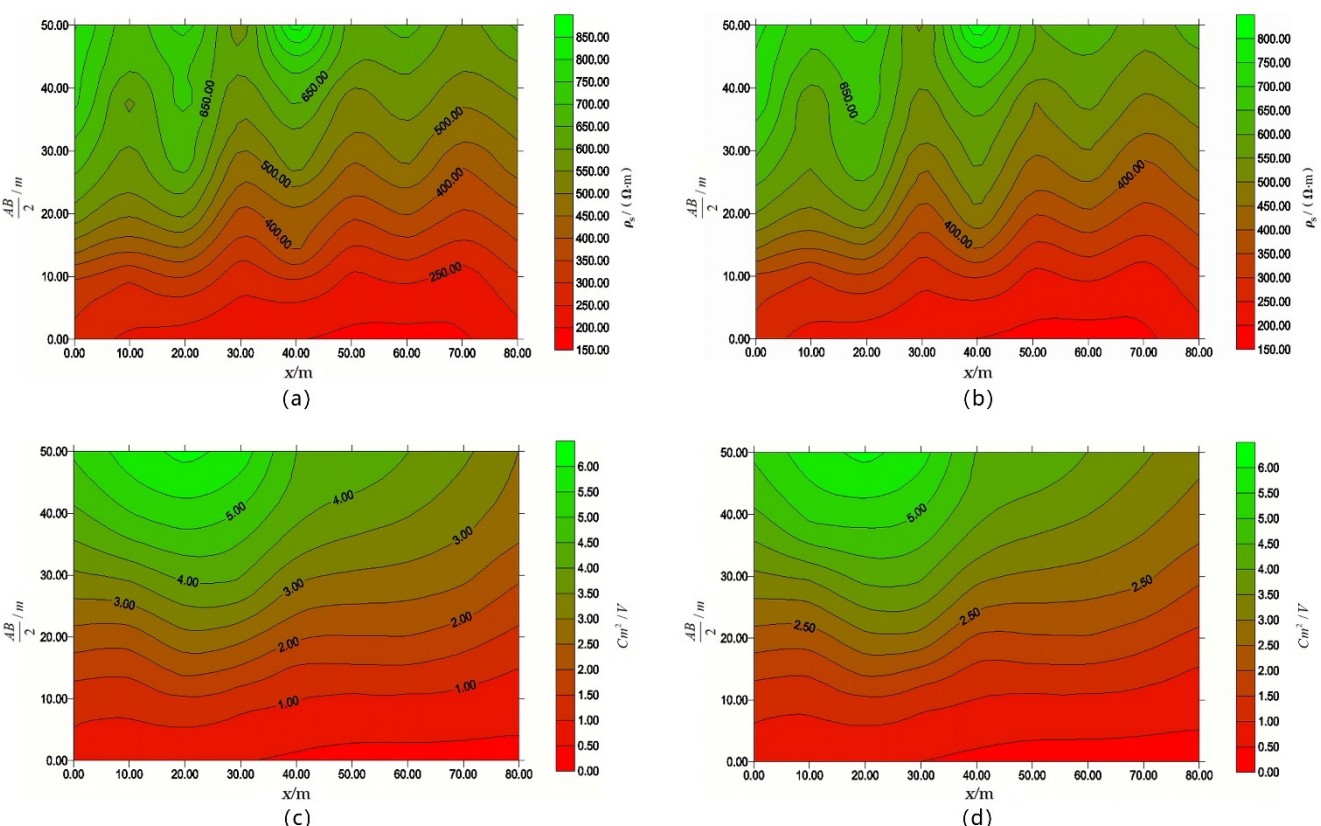

**Figure 9: Pseudo-sections of field comparative experiment. (a)Pseudo-section of apparent resistivity by DUK-2A. (b)Pseudo-section of apparent resistivity by CUGB-CS48DAS. (c)Pseudo-section of apparent polarizability by DUK-2A. (d)Pseudo-section of apparent polarizability by CUGB-CS48DAS.**

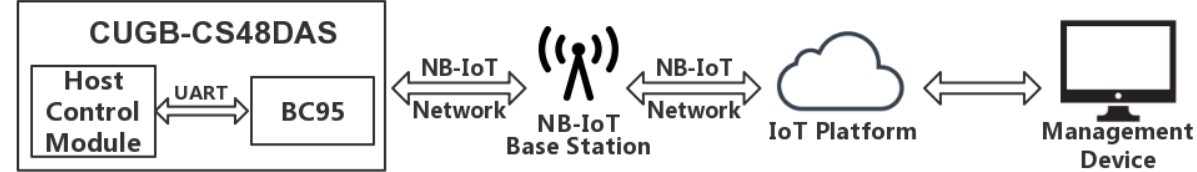

**Figure 10: Connection to the Internet of Things**



| Station Number | Number of Channels | Signal Intensity/dBm | Operational Status | GPS Coordinates | Sampling Rate/kHz | Operational Temperature/℃ | Dropped Packets |
|---|---|---|---|---|---|---|---|
| CUGB48-1 | 48 | -85 | Acquiring | E: 116.351891, N: 39.989734 | 1 | 36.2 | 0 |
| CUGB48-2 | 48 | -86 | Acquiring | E: 116.351489, N: 39.991153 | 1 | 37.1 | 0 |
| CUGB48-3 | 48 | -85 | Acquiring | E: 116.348388, N: 39.991083 | 1 | 35.8 | 0 |
| CUGB48-4 | 24 | -84 | Standing By | E: 116.343493, N: 39.990988 | 1 | 36.5 | 0 |
| CUGB48-5 | 24 | -87 | Standing By | E: 116.348491, N: 39.989098 | 1 | 36.1 | 0 |

**Figure 11: QC Monitoring Interface**