# Peer review of "Development of a New Centralized Data Acquisition System for Seismic Exploration"

_Geoscientific Instrumentation, Methods and Data Systems, 2019_

## Referee Comment (RC1) · Anonymous Referee #1 · 18 Oct 2019

This manuscript gave a detailed description of a centralized seismograph. As is introduced in the manuscript, extensional functions of electrical data acquisition and NB-IoT had also been added to improve the functionality and novelty. In fact, a single piece of equipment that can deal with seismic and electrical data acquisition is difficult to find. Thus, the proposed instrument seems to be a good choice of joint exploration in engineering and geological prospecting. Besides, the result of field test was also provided to show the performance of the proposed instrument in electrical data acquisition. However, there are still a couple of technical points that should be refined and there are also several spelling and writing mistakes need to be corrected. Given that, I recommend that the manuscript under review could be accepted after minor revision.

Details of data transmission speed should be given in subsection 4.4. And in that case,

does it possible for real-time data transmission?

Details of the 5 test points should be given in subsection 4.4. According to the current description, it is not clear where those instruments are placed. Because the signal intensity must be different if the instrument is placed in varies kinds of environments, thus the effect of real-time communication could not be the same.

p. 12, line 25, the sentence "CUGB-CS48DAS can solve the ambiguity problem in geophysical prospecting and achieve joint geophysical seismic and electrical prospecting." is suggested to express as "CUGB-CS48DAS can solve the ambiguity problem in geophysical prospecting by implementing joint seismic and electrical exploration."

The paragraph of 6 Conclusion (4) is hard to follow. Clearer description and analysis are expected to use.

P. 8, line 7, the sentence of "The acquisition systems should be able meet project performance requirements." is supposed to correct as "The acquisition systems should be able to meet project performance requirements."

P. 9, line 30, the 8(d) is mistakenly labeled, and it should be replaced by 9(d).

P. 8, line 29, the explosives should be seismic sources rather than a seismic source.

P. 10, line 31, "on campus" is supposed to be used instead of "in the campus".

P. 15, line 4, the font of "Internal structure" is not in accordance with that of the entire manuscript.

Please also note the supplement to this comment:
https://www.geosci-instrum-method-data-syst-discuss.net/gi-2019-26/gi-2019-26-RC1-supplement.pdf

---

## Referee Comment (RC2) · Anonymous Referee #2 · 24 Feb 2020

Generally speaking, the manuscript described the system structure from both hardware and software, including the core technologies. As for the multiple use of the mentioned instrument, details of various application cases have also been given in the article. To be noted that, comparison with instruments found in the market has also implemented so it is clear to see the advantages of the CUGB-CS48DAS. However, there are still some shortcomings that need to be improved and revised. Given that, I recommend that this article should be accepted after a minor revision.

Details of revisions are as follows:

First of all, the abstract need to be revised to emphasize the technical improvements rather than giving detailed performance inspectors, since they are given in the article and the inspector appeared in the abstract is not a huge technical improvement. So I

[Figure]

recommend that the abstract is supposed to revise to highlight the key technology or the main advantage of the mentioned instrument.

As is described in 4.2 "Realization of Distributed Seismic Exploration Instrument", the CUGB-CS48DAS can be extended to 96 channels by connecting with additional distributed acquisition units with PoE. Is it possible for a single CUGB-CS48DAS to connect with another CUGB-CS48DAS?

P3, "Basic performance parameters and functions of the acquisition system depend on the hardware circuit, which is essential for ensure the quality of the acquired data." the word "ensure" should be "ensuring".

P8, "The acquisition systems should be able meet project performance requirements." is supposed to be "The acquisition systems should be able to meet project performance requirements."

P9, "The Figure 9(c) and 8(d) are illustrations of apparent polarizability." is supposed to be corrected as "The Figure 9(c) and 9(d) are illustrations of apparent polarizability."

---

## Author Response (AR1)

RC: Reviewer's Comment; AR: Authors' Repley; MS: Revision to the Manuscript

**1.1 Response to Reviewer 1**

5   RC: Details of data transmission speed should be given in subsection 4.4. And in that case, does it possible for real-time data transmission?

AR: In order to evaluate the real transmission speed, we uploaded the acquired data file using each of the CUGB-CS48DAS in Figure 11. Because we think upload transmission speed is more important than the download speed. The average transmission speed turned out to 192.32kbps, 206.53kbps, 189.84kbps, 182.06kbps, 226.50kbps respectively. And in this case,

10  real-time acquisition data transmission is actually not available, because theoretically the CUGB-CS48DAS record 140kB of data in a second with a sampling rate of 1kHz.

MS: 4.4, 3$^{rd}$ paragraph

As we can see, a screenshot of QC monitoring interface during a communication experiment we implemented is shown in Figure 11. Totally, 5 test points on campus was selected to observe the network quality and communication speed. CUGB48-

15  1 was placed under a tree with relatively lighter occlusion. CUGB48-2 was placed near a big statue on the grass. CUGB48-3 was placed in the bush near a road. CUGB48-4 was placed in the laboratory near the window. And the last CUGB48-5 was placed right in an open area. We used the indicator Reference Signal Receiving Power (RSRP) to represent the signal intensity as is shown in Figure 11. Generally speaking, the RSRP of 5 test points are around -85dBm which stand for good points of network quality. We then uploaded the acquired data file using each of the CUGB-CS48DAS stations to evaluate

20  the average data uploading speed, and it turned out to be 192.32kbps, 206.53kbps, 189.84kbps, 182.06kbps, 226.50kbps respectively. Under such circumstances, QC monitoring and acquisition control can be realized as is designed.

RC: Details of the 5 test points should be given in subsection 4.4. According to the current description, it is not clear where

25  those instruments are placed. Because the signal intensity must be different if the instrument is placed in varies kinds of environments, thus the effect of real-time communication could not be the same.

AR: CUGB48-1 was placed under a tree with relatively lighter occlusion. CUGB48-2 was placed near a big statue on the grass. CUGB48-3 was placed in the bush near a road. CUGB48-4 was placed in the laboratory near the window. And the last CUGB48-5 was placed right in an open area.

30  MS: 4.4, 3$^{rd}$ paragraph

As we can see, a screenshot of QC monitoring interface during a communication experiment we implemented is shown in Figure 11. Totally, 5 test points on campus was selected to observe the network quality and communication speed. CUGB48-1 was placed under a tree with relatively lighter occlusion. CUGB48-2 was placed near a big statue on the grass. CUGB48-3

was placed in the bush near a road. CUGB48-4 was placed in the laboratory near the window. And the last CUGB48-5 was placed right in an open area. We used the indicator Reference Signal Receiving Power (RSRP) to represent the signal intensity as is shown in Figure 11. Generally speaking, the RSRP of 5 test points are around -85dBm which stand for good points of network quality. We then uploaded the acquired data file using each of the CUGB-CS48DAS stations to evaluate the average data uploading speed, and it turned out to be 192.32kbps, 206.53kbps, 189.84kbps, 182.06kbps, 226.50kbps respectively. Under such circumstances, QC monitoring and acquisition control can be realized as is designed.

RC: P 12, line 25 the sentence "CUGB-CS48DAS can solve the ambiguity problem in geophysical prospecting and achieve joint geophysical seismic and electrical prospecting" is suggested to express as "CUGB-CS48DAS can solve the ambiguity problem in geophysical prospecting by implementing joint seismic and electrical exploration"

AR: The sentence is revised to "CUGB-CS48DAS can solve the ambiguity problem in geophysical prospecting by implementing joint seismic and electrical exploration" as suggested.

MS: 6 Conclusion (4)

(4) CUGB-CS48DAS can solve the ambiguity problem in geophysical prospecting by implementing joint seismic and electrical exploration. As the single method of present geophysical prospecting has ambiguity problems and thus cannot be refined. Therefore, this new centralized data collection system is proposed to provide technical means of solving ambiguity problems while offering useful exploration for joint geophysical prospecting. As the result, the proposed CUGB-CS48DAS becomes a novel supporting equipment with high synchronization precision and acquisition accuracy for joint geophysical seismic and electrical prospecting.

RC: The paragraph of 6 Conclusion (4) is hard to follow. Clearer description and analysis are expected to use.

AR: The paragraph is revised as "CUGB-CS48DAS can solve the ambiguity problem in geophysical prospecting and achieve joint geophysical seismic and electrical prospecting. As the single method of present geophysical prospecting has ambiguity problems and thus cannot be refined. Therefore, this new centralized data collection system is proposed to provide technical means of solving ambiguity problems while offering useful exploration for joint geophysical prospecting. As the result, the proposed CUGB-CS48DAS becomes a novel supporting equipment with high synchronization precision and acquisition accuracy for joint geophysical seismic and electrical prospecting."

MS: 6 Conclusion (4)

(4) CUGB-CS48DAS can solve the ambiguity problem in geophysical prospecting by implementing joint seismic and electrical exploration. As the single method of present geophysical prospecting has ambiguity problems and thus cannot be refined. Therefore, this new centralized data collection system is proposed to provide technical means of solving ambiguity problems while offering useful exploration for joint geophysical prospecting. As the result, the proposed CUGB-CS48DAS

becomes a novel supporting equipment with high synchronization precision and acquisition accuracy for joint geophysical seismic and electrical prospecting.

70

**RC:** P. 8, line 7, the sentence of "The acquisition systems should be able meet project performance requirements." is supposed to correct as "The acquisition systems should be able to meet project performance requirements."

**AR:** The sentence is revised as "The acquisition systems should be able to meet project performance requirements."

75 **MS: 4 Use cases,** 2nd paragraph

The porting of the acquisition system is accomplished in three steps. (1) The acquisition systems should be able to meet project performance requirements. (2) In terms of hardware, the circuit board size and the layout of device connectors and modules should be adjusted. (3) In terms of software, the process controller interfaces should be compiled with the interactive interface, methods manager, document manager, and drawing module.

80

**RC:** P. 9, line 30, the 8(d) is mistakenly labeled, and it should be replaced by 9(d).

**AR:** The mistake has been corrected.

**MS:** 4.3, 2nd paragraph

85 Field comparative experiment was then implemented at the suburban area of Beijing, in contrast with the high-density electrical prospecting instrument of Chongqing Instrument Factory named DUK-2A. The pseudo-sections of apparent resistivity and apparent polarizability is shown as Figure 9. As is indicated in Figure 9(a) and 9(b), the pseudo-sections of apparent resistivity obtained by both instruments present high consistency, no matter in the trend that resistivity varies or the approximate appearance and thickness of different layers. It could also be found that there are some differences between

90 Figure 9(a) and (b) presented as small scaled anomalies. This is because the two instruments were not placed at the exactly same location and there were some errors introduced during inversion process. The Figure 9(c) and 9(d) are illustrations of apparent polarizability. According to that, it is easy to find the contours of apparent polarizability acquired by proposed instrument are well superimposed with that obtained using DUK-2A.

95

**RC:** P. 8, line 29, the explosives should be seismic sources rather than a seismic source.

**AR:** The expression has been modified as suggested.

**MS:** 4.1 2nd paragraph

The instrument was used for a two-week seismic exploration in an area of Golmud City, Qinghai Province in 2013; the total

100 measuring line length was 4.4 km, the seismic migration profile is shown in Figure 6. The exploration adopted the CDJ-Z/P60 geophone of the Chongqing Instrument Factory (technical parameters are listed in Appendix A) and explosives as

seismic sources. The sampling rate was 4 kHz, with 2048 sampling points and 24 channels. Seismic surveying utilizes the coverage measurement method.

105

**RC:** P. 10, line 31, "on campus" is supposed to be used instead of "in the campus".

**AR:** The expression has been modified as suggested.

**MS:** 4.4, 3$^{rd}$ paragraph

As we can see, a screenshot of QC monitoring interface during a communication experiment we implemented is shown in
110 Figure 11. Totally, 5 test points on campus was selected to observe the network quality and communication speed. CUGB48-1 was placed under a tree with relatively lighter occlusion. CUGB48-2 was placed near a big statue on the grass. CUGB48-3 was placed in the bush near a road. CUGB48-4 was placed in the laboratory near the window. And the last CUGB48-5 was placed right in an open area. We used the indicator Reference Signal Receiving Power (RSRP) to represent the signal intensity as is shown in Figure 11. Generally speaking, the RSRP of 5 test points are around -85dBm which stand for good
115 points of network quality. We then uploaded the acquired data file using each of the CUGB-CS48DAS stations to evaluate the average data uploading speed, and it turned out to be 192.32kbps, 206.53kbps, 189.84kbps, 182.06kbps, 226.50kbps respectively. Under such circumstances, QC monitoring and acquisition control can be realized as is designed.

120 **RC:** P. 15, line 4, the font of "Internal structure" is not in accordance with that of the entire manuscript.

**AR:** The font has been modified.

**MS:** Caption of **Figure 3**

Figure 3: Internal structure of CUGB-CS48DAS and circuit boards. (a) Packaging structure of circuit boards. (b) Realize of 5 the Power Supply Board. (c) Realized of the Acquisition Board with 12 channels.

125

**1.2 Response to Reviewer 2**

**RC:** The abstract need to be revised to emphasized the technical improvements rather than giving detailed performance inspectors, since they are given in the article and the inspectors appeared in the abstract is not a huge technical improvement.
130 So I recommend that the abstract is supposed to revise to highlight the key technology or the main advantage of the mentioned instrument.

**AR:** The abstract is revised according to the reviewer's suggestion with key technology and advantages added. Meanwhile, some of the detailed performance inspectors are also deleted.

**MS:** Abstract

135 Seismic exploration equipment has developed rapidly over the past few decades. One such piece of equipment is a centralized seismograph, which plays an important role in engineering, so improving its performance is of great scientific significance. In this research, the core part of general seismic data acquisition devices is packaged to develop a centralized seismic data acquisition system (Named as CUGB-CS48DAS) that has independent operating ability and high scalability, which can be used for seismic exploration in varies engineering usage. Furthermore, by extending and modifying the

140 acquisition circuit and corresponding software, the function of electrical method data acquisition has also been achieved. Thus, the proposed CUGB-CS48DAS makes it possible for joint exploration of seismic and electrical data in a single acquisition station, which is implicitly of great convenience in engineering prospecting as well as a solution to reduce the ambiguity problem. The low-power-consumption computer of the system comprises a 24-bit $\Sigma$-$\triangle$ modulation A/D converter and 48 sampling channels with an optional sampling rate of 50 Hz to 64 kHz. With regard to the host computer, the

145 architecture of the control software is smart, and it can integrate the multiple functions of data acquisition, preprocessing, and self-testing. To complete the networking ability and remote monitoring of this proposed system, the technology of Narrow Band Internet of Things (NB-IoT) was introduced and tested. Field experiments was implemented to prove that the system is stable and convenient to use, and the performance could meet the demand of high-precision joint exploration.

150

**RC:** As is described in 4.2 "Realization of Distributed Seismic Exploration Instrument", the CUGB-CS48DAS can be extended to 96 channels by connecting with additional distributed acquisition units with PoE. Is it possible for a single CUGB-CS48DAS to connect with another CUGB-CS48DAS?

**AR:** Technically a proposed CUGB-CS48DAS can connect with another single instrument to extend acquisition traces to 96,

155 but we do not think that it's an economic way of extending the number of traces. Thus, this function is not realized yet.

**MS:** No revision.

**RC:** P3 "Basic performance parameters and functions of the acquisition system depend on the hardware circuit, which is essential for ensure the quality of the acquired data" the word "ensure" should be "ensuring".

160 **AR:** Corrected as suggested.

**MS: 2 Structure of the Acquisition System,** 1st paragraph

The development of the acquisition system generally involves hardware circuit design and control software development (D.

10 Marjanovic et al., 2015). The CUGB-CS48DAS system structure is shown in Figure 2. Basic performance parameters and functions of the acquisition system depend on the hardware circuit, which is essential for ensuring the quality of the acquired

165 data. The operating state of the overall system is coordinated by the software, which is subject to certain fault tolerance and more flexible than the hardware circuit. It is required to encapsulate operations near hardware bottom and provide flexible and safe interfaces, thus both ensuring the system stability and improving its flexibility and portability.

170 **RC:** P8, "The acquisition systems should be able meet project performance requirements" is supposed to be "The acquisition systems should be able to meet project performance requirements"

**AR:** The sentence is corrected as "The acquisition systems should be able to meet project performance requirements".

**MS:** 4 Use Cases, 2nd paragraph

The porting of the acquisition system is accomplished in three steps. (1) The acquisition systems should be able to meet

175 project performance requirements. (2) In terms of hardware, the circuit board size and the layout of device connectors and modules should be adjusted. (3) In terms of software, the process controller interfaces should be compiled with the interactive interface, methods manager, document manager, and drawing module.

**RC:** P9, "The Figure 9(c) and 8(d) are illustrations of apparent polarizability." is supposed to be corrected as "The Figure 9(c)

180 and 9(d) are illustrations of apparent polarizability."

**AR:** The incorrect label has been corrected to 9(d), and another incorrect label has also been corrected.

**MS:** 4.3, 2nd paragraph

[revised manuscript text omitted]

Qc  Search  Connect  Begin  Stop  Location  States

745

**Figure 11: QC Monitoring Interface.**

---

## Referee Report (RR1)

Dear editor:

The idea of this manuscript is original and unique, and it describes the technical realization of a new centralized data acquisition system for geophysical exploration. The technique mentioned is very novel. Through the authors' revisions, the expression is more fluent and understandable. The paper is probably publishable. I recommend the acceptance of the paper as it is.

---

## Author Response (AR2)

RC: Reviewer's Comment; AR: Authors' Repley; MS: Revision to the Manuscript

**Response to Mr. Jean Dumoulin**

Dear editor

First of all, I should thank you for reviewing our manuscript and your precious suggestion. And according to the comments, our responses are as follow:

RC: Some indications about energy consumption of the whole system would be of interest.

AR: We measured the different currents with a multimeter after each DC-DC to evaluate the system power consumption. In standby mode, which is the most common state, the power consumption is 2.065W. In working condition, each acquisition board reaches the maximum power consumption of 2.66W at the highest sampling rate 64ksps. When all 4 acquisition boards (48 channels) work in the same time, the system power consumption reaches 12.705W, which is also the maximum power consumption of the system.

The system we designed is powered by external 12V power source, therefore, there are different ways to solve the energy problem in the field. A single station can be powered by a custom made 12V output lithium battery that is sustainable in 15 hours. The other way is to utilize the power station we designed to supply power to multiple stations simultaneously. In this way, the power supply is more stable than using the battery directly.

MS: 5.1, a new subtitle is added:

**5.1 System Power Consumption**

We measured the different currents with a multimeter after each DC-DC to evaluate the system power consumption. In standby mode, which is the most common state, the power consumption is 2.065W. In working condition, each acquisition board reaches the maximum power consumption of 2.66W at the highest sampling rate 64ksps. When all 4 acquisition boards (48 channels) work in the same time, the system power consumption reaches 12.705W, which is also the maximum power consumption of the system.

The system we designed is powered by external 12V power source, therefore, there are different ways to solve the energy problem in the field. A single station can be powered by a custom made 12V output lithium battery that is sustainable in 15 hours. The other way is to utilize the power station we designed to supply power to multiple stations simultaneously. In this way, the power supply is more stable than using the battery directly.

**List of changes**

1. A new subtitle named **5.1 System Power Consumption** is added to explain the power consumption of the system and to provide power solution furthermore.

[revised manuscript text omitted]